# Urinary Phenolic Metabolites Associated with Peanut Consumption May Have a Beneficial Impact on Vascular Health Biomarkers

**DOI:** 10.3390/antiox12030698

**Published:** 2023-03-11

**Authors:** Isabella Parilli-Moser, Inés Domínguez-López, Anna Vallverdú-Queralt, Sara Hurtado-Barroso, Rosa M. Lamuela-Raventós

**Affiliations:** 1Department of Nutrition, Food Sciences and Gastronomy, School of Pharmacy and Food Sciences XIA, Institute of Nutrition and Food Safety (INSA-UB), University of Barcelona, 08028 Barcelona, Spain; 2CIBER Fisiopatología de la Obesidad y Nutrición (CIBEROBN), Instituto de Salud Carlos III, 28029 Madrid, Spain; 3Department of Medicine, School of Medicine and Health Sciences, University of Barcelona, 08036 Barcelona, Spain

**Keywords:** antioxidant, antiplatelet, antithrombotic, eicosanoids, polyphenols, vascular health

## Abstract

Phenolic compounds in peanuts may moderate inflammation and endothelial function. Thus, the aim of this study was to evaluate the association of urinary phenolic metabolites (UPMs) with vascular biomarkers after peanut product consumption. A three-arm parallel-group randomized controlled trial was conducted in 63 healthy young adults who consumed 25 g/day of skin roasted peanuts (SRP), 32 g/day of peanut butter (PB), or 32 g/day of a control butter for six months. UPMs were analyzed by liquid chromatography coupled to mass spectrometry. Additionally, urinary eicosanoids, prostacyclin I2 (PGI_2_), and thromboxane A2 (TXA_2_) were determined using two competitive enzyme-linked immunosorbent assay kits. Consumers of SRP and PB presented significantly higher excretion of UPMs (enterodiol glucuronide (*p* = 0.018 and *p* = 0.031), 3-hydroxybenzoic acid (*p* = 0.002 and *p* < 0.001), vanillic acid sulfate (*p* = 0.048 and *p* = 0.006), *p*-coumaric acid (*p* = 0.046 and *p* = 0.016), coumaric acid glucuronide I (*p* = 0.001 and *p* = 0.030) and II (*p* = 0.003 and *p* = 0.036), and isoferulic acid (*p* = 0.013 and *p* = 0.015) in comparison with the control group. An improvement in PGI_2_ (*p* = 0.037) levels and the TXA_2_:PGI_2_ ratio (*p* = 0.008) was also observed after the peanut interventions compared to the control. Interestingly, UPMs with significantly higher post-intervention levels were correlated with an improvement in vascular biomarkers, lower TXA_2_ (r from −0.25 to −0.48, *p* < 0.050) and TXA_2_:PGI_2_ ratio (r from −0.25 to −0.43, *p* < 0.050) and higher PGI_2_ (r from 0.24 to 0.36, *p* < 0.050). These findings suggest that the UPMs with higher excretion after peanut product consumption could have a positive impact on vascular health.

## 1. Introduction

The regular consumption of nuts and peanuts has been associated with a reduced risk of developing cardiovascular diseases [1,2,3] and diabetes [4,5,6], with improvements in the lipid profile, inflammation markers, and preservation of endothelial function [7,8,9,10]. However, the results of studies evaluating the impact of nut consumption on inflammation are discrepant, as clinical trials have not been able to consistently verify the anti-inflammatory effects found in observational studies [8,11,12]. 

Peanuts are edible seeds classified as legumes, nevertheless, they are frequently include in the nuts group, since they share a similar nutritional composition, being nutrient-dense and rich in monounsaturated fatty acids [13,14]. They are the most consumed nuts worldwide [15], and are regarded as a convenient, tasty, and easy snack that contributes to a healthy lifestyle [14]. The wide range of nutrients and bioactive compounds found in peanuts include fiber, folate, and arginine [13,16], and also, they are a well-known source of antioxidants such as polyphenols, whose concentration have been reported to be highest in their skins [17,18]. Phenolic acids, mainly *p*-coumaric and isoferulic acids, were the most abundant polyphenols found in skin roasted peanuts and peanut butter with skin added, representing more than 60–70% of the total polyphenols [19].

Polyphenols represent the main antioxidants and anti-inflammatory compounds in our diet and have been related to antithrombotic and vasodilatory effects [20]. They are also reported to play a beneficial role in the prevention of inflammation-related chronic diseases such as type 2 diabetes, obesity, cancers, and neurodegenerative and cardiovascular diseases [21,22,23]. Phenolic compounds appear to regulate the expression of several pro- and anti-inflammatory genes and cytokines through MAPK, NF-kB, and arachidonic acid pathways, contributing to the inhibition of enzymes involved in eicosanoid production and enhancing anti-inflammatory activities [24,25]. However, the role of polyphenols in modulating inflammatory pathways needs further investigation. It is believed that the immunoprotective and anti-inflammatory activities of polyphenols are initiated in the gut, with subsequent systemic effects [21]. 

The eicosanoids prostacyclin I_2_ (PGI_2_) and thromboxane A2 (TXA_2_) are the major arachidonic acid products in the vascular endothelium and platelets, respectively [26,27,28]. TXA_2_ has prothrombotic and vasoconstrictor properties, as it stimulates inflammation and platelet aggregation [28]. In contrast, PGI_2_ acts as a potent vasodilator and inhibitor of platelet aggregation [26,29], counteracting the activities of TXA_2_ and playing an important role in preventing atherosclerosis and thrombosis [28,30]. Despite being well-known vascular biomarkers, the association of PGI_2_ and TXA_2_ with peanut consumption and urinary phenolic metabolites (UPMs) has not been studied to date. Thus, the aim of the present study was to evaluate UPMs’ concentrations after daily intake of peanuts or peanut butter and their potential effect on vascular health through the analysis of urinary eicosanoids.

## 2. Materials and Methods

### 2.1. Study Population and Recruitment

Healthy young adults aged 18–33 years were recruited into the ARISTOTLE study from the Food and Nutrition Torribera Campus at the University of Barcelona and the surrounding area through poster boards in different settings, flyer distribution, and word of mouth. Potential participants were screened using the following exclusion criteria: body mass index (BMI) over 25 kg/m^2^, history of chronic diseases (cardiovascular diseases, cancer, diabetes, and others), peanut allergy, active smoking, excessive alcohol consumption, and other toxic habits.

### 2.2. Study Design 

The present study includes data from a three-arm randomized controlled trial (ARISTOTLE study), described elsewhere [31]. All participants signed an informed consent form and were randomized to one of three intervention groups, consuming either 25 g/day of skin roasted peanuts (SRP) or two tablespoons (32 g)/day of peanut butter (PB) or two tablespoons (32 g)/day of a control butter (CB) based on peanut oil and free of fiber and polyphenols. Prior to the baseline visit they followed a two-week peanut-free run-in period. The intervention lasted 6 months, extended in some cases to 7 months due to the COVID-19 pandemic. To facilitate the intervention compliance, the participants were supplied with the three intervention products and requested to follow their habitual diet, excluding wine, grapes, dark chocolate (>70% cocoa), berries, and nuts. 

The study was conducted in compliance with the principles of the Declaration of Helsinki. Ethical approval for the involvement of human subjects was granted by the Ethics Committee of Clinical Investigation of the University of Barcelona (Barcelona, Spain) and the clinical trial was registered at https://register.clinicaltrials.gov (NCT04324749). 

### 2.3. Measurements and Outcome Assessment 

#### 2.3.1. Anthropometric and Clinical Measurements 

Anthropometric and clinical measurements were obtained in fasting conditions at the beginning and end of the trial. Height was measured in the standing position using a portable stadiometer. Weight and body composition (body fat and muscle percentages) were measured using a tetrapolar OMRON BF511 bioelectrical device, with the participants wearing light clothes and no shoes. BMI was calculated as weight divided by height squared (kg/m^2^). Using an inelastic flexible tape, waist circumference was measured at the midpoint between the lower margin of the last rib and the top of the iliac crest, and hip circumference on the upper trochanters. Both measurements were used to calculate the waist-to-hip ratio, dividing waist circumference by hip circumference. Diastolic and systolic blood pressure (DBP and SBP, respectively) were measured in triplicate using an OMRON M6 digital monitor with the volunteer in a sitting position. 

#### 2.3.2. Sample Collection and Biochemical Analysis 

Blood and urine samples were collected at baseline and at the end of the intervention. Overnight fasting blood was obtained from the arm via venipuncture into tubes containing ethylenediaminetetraacetic acid (EDTA) to separate serum after centrifugation at 3000 g for 10 min at 4 °C. Urine from 24 h before each visit was provided by participants. All samples were aliquoted and stored at −80 °C until analysis. Biochemical markers in serum (lipid profile) were measured in an external laboratory (Cerba international, Barcelona, Spain) using enzymatic methods. 

#### 2.3.3. Dietary Intake and Physical Activity 

Diet and physical activity were recorded by professional staff members using validated questionnaires. Dietary intake was quantified using a semi-quantitative 151-item food frequency questionnaire (FFQ) and Spanish food composition tables [32]. Physical activity was measured as the metabolic equivalent of task-minutes per week (MET/week) using the Spanish version of the Minnesota Leisure-Time Physical Activity Questionnaire [33]. 

### 2.4. Urinary Phenolic Metabolites Analysis

#### 2.4.1. Standards and Reagents

Protocatechuic acid, 4-hydroxybenzoic acid, *o*-coumaric acid, *m*-coumaric acid, *p*-coumaric acid, enterodiol, urolithin-A, and urolithin-B were purchased from Sigma-Aldrich (St. Louis, MO, USA). Dihydroresveratrol and the internal standard (+) cis, trans-abscisic acid D6 were obtained from Santa Cruz (Santa Cruz Biotechnology, Santa Cruz, CA), and 3-hydroxybenzoic acid, vanillic acid, syringic acid, and enterolactone from Fluka (St. Louis, MO, USA). The reagents were purchased from the following commercial suppliers: methanol and acetonitrile of HPLC grade from Sigma-Aldrich, formic acid (≥98%) from Panreac Química S.A. (Barcelona, Spain), and ultrapure water (Milli-Q) generated by a Millipore system (Bedford, MA, USA).

#### 2.4.2. Urine Treatment for Phenolic Metabolite Analysis

In each visit, 24 h urine samples were collected and stored at −80 °C until analysis. All samples and standards were always handled under filtered light and cool conditions to prevent phenolic oxidation. UPMs were determined following a validated method developed by our research group [34]. Briefly, 50 μL of urine was diluted with ultrapure water to 1 mL, acidified with 2 μL of formic acid and centrifuged at 15,000 g at 4 °C for 4 min. The acidified urines underwent a solid-phase extraction in Water Oasis HLB 96-well plates (30 µm) (Water Oasis, Milford, MA, USA). The 96-well plate was activated with 1 mL of methanol and 1 mL of 1.5 M formic acid, added consecutively. Then, 1 mL of sample was loaded onto the plates together with 100 μL of the internal standard. Sample clean-up was performed with 500 μL of 1.5 M formic acid and 0.5% methanol, and elution was achieved using 1 mL of methanol acidified with 1.5 M formic acid. After evaporation under nitrogen stream, it was reconstituted with 100 μL of 0.05% formic acid and the extract was filtered with a 0.22 µm polytetrafluoroethylene 96-well plate filter (Millipore, MA, USA). 

#### 2.4.3. Chromatographic Conditions

The analysis was performed by liquid chromatography coupled to linear trap quadrupole Orbitrap high-resolution mass spectrometry (LC-LTQ-Orbitrap-HRMS) (Thermo Scientific, Hemel Hempstead, UK) equipped with electrospray ionization and working in negative mode, as previously described by Laveriano-Santos et al. [34]. Chromatographic separation was performed using a Kinetex F5 100A (50 × 4.6 mm × 2.6 µm) from Phenomenex (Torrance, CA, USA). The gradient elution was performed with two mobile phases, A, water (0.05% formic acid), and B, acetonitrile (0.05% formic acid), using the following non-linear gradient: 0 min, 2% B; 1 min, 2% B; 2.5 min, 8% B; 7 min, 20% B; 9 min, 30% B; 11 min, 50% B; 12 min, 70% B; 15 min, 100% B; 16 min, 100% B; 16.5 min, 2% B; and 21.5 min, 2% B. The flow rate was set at 0.5 mL/min and the injection volume was 5 µL. 

#### 2.4.4. Identification and Quantification of Urinary Phenolic Metabolites

Aglycones were identified by comparing retention times with those of available standards and phase II metabolites by comparison with accurate mass MS/MS spectra with an error of 5 ppm found in the literature. As standards for glucuronidated and sulfated UPMs were unavailable, these metabolites were quantified with their respective aglycone equivalents. Xcalibur 3.0 and Trace Finder version 4.1 (Thermo Fisher Scientific, San Jose, CA, USA) software were used for the instrument control and chromatographic data analysis. In this study, 38 UPMs were identified and quantified (aglycones and phase II metabolites in glucuronide and sulfate form). Values below the limit of detection were replaced by half the limit of detection, and values below the limit of quantitation were replaced by the midpoint between the limit of detection and the limit of quantitation.

### 2.5. Determination of Eicosanoids in Urine 

The concentration of urinary PGI_2_ and TXA_2_ was indirectly quantified by measuring the prostaglandin I metabolite and 11-dehydro thromboxane B2, respectively. Both molecules were determined in urine using two competitive enzyme-linked immunosorbent assay (ELISA) kits acquired from Cayman Chem. Co. (Ann Arbor, MI, USA, ref. 501,100 and 519,510). The PGIM assay has a range from 39 to 5000 pg/mL and a sensitivity (80% B/B0) of approximately 120 pg/mL. The 11-dehydro thromboxane B2 assay has a range from 15.6 to 2000 pg/mL and a sensitivity (80% B/B_0_) of approximately 34 pg/mL. The urine samples were diluted 1:10 and 1:3, respectively, and assayed in triplicate. The TXA_2_:PGI_2_ ratio was also calculated. Concentrations are expressed as pg/mL.

### 2.6. Statistical Analyses 

Continuous variables are expressed as mean ± standard deviation (SD) and categorical variables are expressed as number (n) and proportion (%). Normality of distribution was assessed by the Shapiro–Wilk test. Non-parametric tests were used due to the non-normality of most variables and the small sample size (<30 in each group). Differences between groups in the general characteristics of participants at baseline were measured by the chi-square test for categorical variables and the Kruskal–Wallis test for continuous variables. The effect of peanut and peanut butter interventions on UPMs and eicosanoids was estimated by performing a generalized estimating equation on Poisson regression models for repeated measures. Identity link function, autoregressive correlation, and robust standard error parameters were specified due to the low number of clusters and the nature of the variables. Analyses were adjusted for age, sex, and physical activity. Finally, Spearman’s correlation coefficient was estimated to study linear associations between UPMs and eicosanoids. All statistical analyses were conducted using the Stata statistical software package version 16.0 (StataCorp, College Station, TX, USA). Differences were considered significant when the *p*-value was lower than 0.050.

## 3. Results

### 3.1. Baseline Characteristics

Among the 90 subjects who enrolled and were randomized to each arm, 63 participants (19 males and 44 females) completed the study. Table 1 shows their general characteristics. The average age was 22.71 ± 3.13 years; around 70% were female and 46% had graduated from a 4-year degree course. The mean physical activity was higher than 4000 METs/week. At baseline, no significant differences between groups were found, except in the level of plasmatic high-density lipoprotein cholesterol (HDL-c) (*p* = 0.006) and urinary concentrations of urolithin B and dihydroresveratrol glucuronide II (*p* = 0.022 and *p* = 0.008 respectively, data not shown). 

### 3.2. Effect of the Intervention on Urinary Phenolic Metabolite Levels 

The concentration of UPMs by polyphenol class (lignans, hydroxybenzoic acids, hydroxycinnamic acids, stilbenes, and hydroxycoumarins) is presented in Table 2. A total of 38 metabolites were identified in urine and many of them were detected in the form of glucuronides and sulfates. Overall, the most abundant UPMs were hydroxybenzoic acids, and the least abundant were stilbenes. After adjustment for sex and age, the excretion of some UPMs was found to be higher after peanut or peanut butter consumption compared with the control butter. 

Compared to the CB group, lignan excretion was significantly higher in the SRP group (enterodiol glucuronide, *p* = 0.018; enterolactone glucuronide, *p* = 0.045; and enterolactone sulfate, *p* = 0.020) and the PB group (enterodiol glucuronide, *p* = 0.031, and enterolactone glucuronide, *p* = 0.032) after full adjustment (Table 2). In regard to the hydroxybenzoic acids, higher excretion levels after the consumption of PB (3-hydroxybenzoic acid, *p* < 0.001; hydroxybenzoic acid sulfate, *p* = 0.014; vanillic acid sulfate, *p* = 0.006; and syringic acid glucuronide II, *p* = 0.023) and SRP (3-hydroxybenzoic acid, *p* = 0.002; vanillic acid sulfate, *p* = 0.048; and syringic acid sulfate, *p* = 0.041) were found compared to the CB group. Interestingly, post-intervention levels of hydroxycinnamic acids such as *p*-coumaric acid, coumaric acid glucuronide I, coumaric acid glucuronide II, and isoferulic acid were significantly higher in both the PB *(p* = 0.016, *p* = 0.030, *p* = 0.036, and *p* = 0.015, respectively) and SRP groups (*p* = 0.046, *p* = 0.001, *p* = 0.003, and *p* = 0.013, respectively) in comparison with the control. Regarding stilbenes, the only increase observed was in dihydroresveratrol glucuronide II after PB consumption versus CB (*p* = 0.004) (models of adjustment are shown in Table A1).

### 3.3. Effect of the Intervention on Eicosanoid Levels in Urine 

The urinary levels of eicosanoids are presented in Table 3. Compared to the control, a significant change in PGI_2_ levels was observed after SRP consumption (*p* = 0.037), whereas the TXA_2_:PGI_2_ ratio decreased after both SRP and PB interventions (*p* = 0.021 and *p* = 0.047, respectively) after adjustment. However, no change in TXA_2_ levels was observed after 6 months or between groups (models of adjustment are shown in Table A2).

### 3.4. Relationship between Urinary Phenolic Metabolites and Eicosanoids 

Correlations were generated to evaluate the association between UPM and eicosanoid levels. Of the 38 quantified metabolites, 17 showed a significant correlation with one or two of the vascular biomarkers (Figure 1). The participants with a higher excretion of enterodiol, enterolactone, enterolactone glucuronide, enterolactone diglucuronide, enterolactone sulfate, syringic acid glucuronide I, syringic acid sulfate, dihydroresveratrol glucuronide II, urolithin A, and urolithin B presented lower levels of TXA_2_ (r = −0.44, *p* < 0.001; r = −0.36, *p* = 0.003; r = −0.25, *p* = 0.045; r = −0.41, *p* < 0.001; r = −0.36, *p* = 0.005; r = −0.31, *p* = 0.015; r = −0.28, *p* = 0.029; r = −0.25, *p* = 0.046; r = −0.48, *p* <0.001; and r = −0.38, *p* = 0.002, respectively). Moreover, significant direct correlations were observed between levels of PGI_2_ and enterodiol glucuronide, 3-hydroxybenzoic acid, vanillic acid, *p*-coumaric acid, coumaric acid glucuronide II, and isoferulic acid (r = 0.26, *p* = 0.045; r = 0.26, *p* = 0.042; r = 0.36, *p* = 0.006; r = 0.27, *p* = 0.032; r = 0.24, *p* = 0.046; and r = 0.31, *p* = 0.014, respectively). Similarly, higher levels of enterodiol, enterolactone, enterolactone sulfate, 3-hydroxybenzoic acid, vanillic acid sulfate, *p*-coumaric acid, o-coumaric acid, coumaric acid glucuronide III, isoferulic acid, dihydroresveratrol glucuronide II, and urolithin A were associated with a lower TXA_2_:PGI_2_ ratio (r = −0.26, *p* = 0.042; r = −0.29, *p* = 0.019; r = −0.28, *p* = 0.023; r = −0.27, *p* = 0.031; r = −0.25, *p* = 0.046; r = −0.27, *p* = 0.038; r = −0.30, *p* = 0.017; r = −0.29, *p* <0.022; r = −0.43, *p* <0.001; r = −0.28, *p* = 0.027; and r = −0.41, *p* = 0.001; respectively).

## 4. Discussion

In this randomized controlled trial, a significant increase in urinary UPMs was observed in healthy young adults who consumed SRP and PB daily for 6 months compared to those who consumed CB (a cream without fiber or polyphenols). Similarly, the ratio between the eicosanoids TXA_2_ and PGI_2_ improved in the consumers of SRP or PB compared to CB. Interestingly, we found that several UPMs with significantly higher post-intervention levels were associated with improvements in vascular biomarkers (lower TXA_2_ and TXA_2_:PGI_2_ ratio and higher PGI_2_). 

Compared to the control group, participants consuming SRP and PB were found to excrete higher levels of lignans (enterodiol glucuronide, enterolactone glucuronide, and enterolactone sulfate), hydroxybenzoic acids (3-hydroxybenzoic acid vanillic acid sulfate, hydroxybenzoic acid sulfate, syringic acid glucuronide II, and syringic acid sulfate), hydroxycinnamic acids (*p*-coumaric acid, coumaric acid glucuronides I and II, and isoferulic acid), and stilbenes (dihydroresveratrol glucuronide II). To date, few studies have investigated the bioavailability of peanut polyphenols. In a recent study published by our group, we showed that the most abundant polyphenols in the two intervention products (SRP and PB) are *p*-coumaric and isoferulic acid [19]. In a comparative study with tree nuts, Rocchetti et al. found the total phenolic content was highest in peanuts, especially phenolic acids such as 3,4-dihydroxyphenylacetic, 4-hydroxybenzoic, and protocatechuic acids, after in vitro fecal fermentation [35]. 

In vitro and in vivo studies have provided evidence for the anti-inflammatory, antiadipogenic, and antidiabetic potential of nut polyphenols [36,37,38]. Nevertheless, the biological properties of these phytochemicals are highly dependent on their bioavailability. After ingestion, 85–90% of dietary polyphenols reach the large intestine, where they become fermentable substrates for bacterial enzymes, leading to the breakdown of their original structures into several smaller absorbable metabolites [39,40,41]. These compounds reach the bloodstream and can have a biological effect on target organs. [39]. Maintaining a healthy gut microbiota has emerged as a key factor for protection against inflammatory-related diseases. Polyphenol activity is thought to principally take place in the gut, where phenolic immunoprotective and anti-inflammatory effects are initiated before acting at a systemic level [21]. Regarding microbial metabolites, participants in the present study who consumed SRP or PB presented a higher excretion of enterolactone glucuronide and enterodiol glucuronide, both important biomarkers of microbiota diversity [42]. In previous studies, higher post-intervention levels of phenolic metabolites have been associated with beneficial health effects. For example, hydroxycinnamic acids were related with lower odds of depression in an Italian cohort [43] and of developing metabolic syndrome in a Polish cohort [44]. Hydroxybenzoic acids were inversely associated with cardiovascular disease in a Spanish study [45]. In addition, associations have also been found between urinary lignan metabolites and a lower risk of type 2 diabetes (enterolactone) and diabetes mortality (enterodiol) [46,47].

The eicosanoids PGI_2_ and TXA_2_ are the major arachidonic acid products in the vascular endothelium and platelets, synthesized by cyclooxygenase isoforms [26,27,28]. As PGI_2_ counteracts the pro-aggregatory and vasoconstrictor activities of TXA_2_ [29,31], the ratio of the two molecules is an important regulator of the interaction between platelets and vessel walls in vivo, and crucial for vascular health [48,49]. Previous research indicates that peanut consumption may have a positive effect on cardiometabolic risk factors and reduce the risk of developing cardiovascular diseases [3,5,50,51]. However, this is the first study to report an improvement in vascular biomarkers related to antithrombotic and vasodilatory effects in healthy young adults after peanut product consumption. We found a significant reduction in the TXA_2_:PGI_2_ ratio in participants who daily consumed SRP and PB compared to the control group. Regarding PGI_2_, a higher level was found in the SRP group and an increasing tendency in PB consumers, whereas no changes were observed in TXA_2_ levels after the intervention compared to the control. Our results agree with those of Canales et al., who reported an increase in PGI_2_ serum levels and a decrease in the TXB_2_:PGI_2_ ratio after consumption of walnut-paste-enriched meat [52]. Similarly, a long-term decrease in inflammatory markers was observed in healthy volunteers after the consumption of 20 g and 50 g of Brazil nuts [53]. 

The role of phenolic compounds in anti-inflammatory reactions and the modulation of enzymatic activities related to eicosanoid synthesis and degradation has been reported [54,55]. They are thought to be involved in the expression of several pro- and anti-inflammatory genes and cytokines through different pathways (MAPK, NF-kB, and arachidonic acid) [24,25]. To shed light on the vascular effects of polyphenols, we correlated UPM with eicosanoid levels and found that participants who excreted more *p*-coumaric acid, *o*-coumaric acid, coumaric acid glucuronide III, and isoferulic acid (the major polyphenols in peanuts) presented higher levels of PGI_2_ and a lower TXA_2_:PGI_2_ ratio. In addition, those who excreted more enterodiol, enterodiol glucuronide, enterolactone, enterolactone sulfate and glucuronide, and urolithins A and B (microbial phenolic metabolites) presented a lower TXA_2_:PGI_2_ ratio and TXA_2_ levels. These results suggest that an improvement in vascular function is associated with a higher excretion of UPMs from a dietary source (in this case peanut consumption) or from the gut microbiota. 

In line with our results, it has been demonstrated that plant polyphenols can enhance PGI_2_ release from endothelial cells [56], and the consumption of high-procyanidin chocolate was related to an increase in plasma PGI_2_ [55]. Additionally, a reduction in serum TXA_2_ was determined in healthy subjects after the consumption of extra virgin olive oil, a typical polyphenol-rich product of the Mediterranean diet [57]. After an intervention with cranberry juice, Rodriguez-Mateos et al. found that twelve polyphenol metabolites, including ferulic and caffeic acid sulfates, quercetin-3-O-ß-D glucuronide, and γ-valerolactone sulfate, were significantly correlated with improved vascular function in healthy volunteers [58]. They also observed an amelioration of endothelial function after the acute intake of blueberry drinks containing different levels of polyphenols [59] and raspberries [60]. Additionally, metabolites such as caffeic acid, ferulic acid, isoferulic acid, vanillic acid, and 2-hydroxybenzoic acid, measured in plasma after the intake of a whole-grain biscuit rich in phenolic acids, were associated with a reduced inflammatory status in overweight subjects [61]. Moreover, after consumption of a low-polyphenol diet by healthy young men, a higher ratio of TXA_2_:PGI_2_ versus a usual diet (Mediterranean diet) was observed [62]. Nevertheless, conflicting results have also been published. For example, no changes in TXA_2_ levels were found in healthy subjects who consumed 40 g of dark chocolate [63]. Moreover, no effects on TXA_2_ and PGI_2_ metabolites in urine, or the ratio of both molecules, were found in healthy subjects consuming an American diet supplemented with procyanidin-enriched cacao [64].

### Strengths and Limitations 

To our knowledge, this is the first study to analyze UPMs after peanut consumption and to provide promising results regarding the effect of peanut consumption on vascular function in healthy young adults. Another strong point of the present study is the randomized and controlled design, as well as the use of a precise extraction of phenolic metabolites from urine samples and the novel method based on liquid chromatography coupled to mass spectrometry used for the accurate identification and quantification of UPMs. However, several limitations should also be acknowledged, including the small sample size for each intervention group and the lack of blinding. The sample size was calculated to ensure 80% of statistical power, but this value decreased to 60% due to dropouts. Finally, the scope of the study did not include the elucidation of molecular mechanisms underlying the observed associations, and hence, causality cannot be determined. 

## 5. Conclusions

In conclusion, the present study shows for the first time that regular peanut and peanut butter consumption could have a positive impact on vascular biomarkers in healthy young adults. Our results suggest that the urinary phenolic metabolites whose production increased after peanut product consumption, especially hydroxycinnamic acids, may contribute to the maintenance of vascular health, as could microbial phenolic metabolites such as enterolignans and hydroxycoumarins. However, further studies, mainly clinical trials, are needed to elucidate the association between metabolites and vascular function, as well as to understand the plausible mechanisms.

## Figures and Tables

**Figure 1 antioxidants-12-00698-f001:**
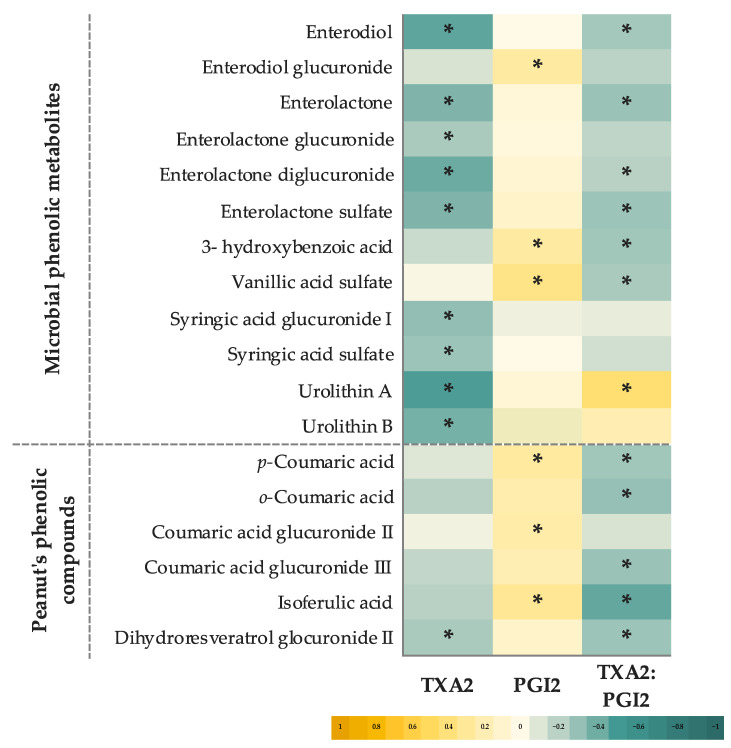
Heatmap of correlations between urinary phenolic metabolites and eicosanoids. The value of the correlation coefficient (r) ranging from −1 to 1 is expressed in green and yellow, respectively, in the bar bellow. *: refers to significant Spearman correlations *p* < 0.005.

**Table 1 antioxidants-12-00698-t001:** General characteristics of the study population at baseline.

	SRP (n = 21)	PB (n = 23)	CB (n = 19)	*p*-Value
**Female, n (%)**	14 (66)	18 (78)	12 (63)	0.528
**Age (years)**	22.28 ± 3.20	23.43 ± 2.90	22.42 ± 3.29	0.247
**Education level, n (%)**				0.512
University students	11 (52%)	11 (48%)	12 (63%)	
Graduated	10 (48%)	12 (52%)	7 (37%)	
**Physical activity (METs/week)**	4850 ± 2124	4703 ± 2381	4607 ± 1728	0.954
**Anthropometric measurements**				
Weight (kg)	63.26 ± 10.12	60.10 ± 7.72	63.78 ± 10.04	0.412
BMI (kg/m^2^)	22.12 ± 3.52	22.19 ± 2.60	22.59 ± 2.67	0.679
Waist circumference (cm)	72.73 ± 8.31	71.28 ± 5.53	74.68 ± 5.99	0.228
Waist to hip ratio	0.74 ± 0.06	0.74 ± 0.05	0.77 ± 0.05	0.130
Body fat (%)	26.66 ± 8.07	28.45 ± 7.88	26.22 ± 7.99	0.628
**Lipid profile**				
TG (mmol/L)	0.71 ± 0.20	0.85 ± 0.35	0.80 ± 0.25	0.341
TC (mmol/L)	4.33 ± 0.52	4.60 ± 0.88	4.09 ± 0.64	0.137
LDL-c (mmol/L)	2.22 ± 0.39	2.60 ± 0.69	2.30 ± 0.50	0.142
HDL-c (mmol/L)	1.75 ± 0.30	1.59 ± 0.53	1.50 ± 0.30	**0.006**
**Blood pressure**				
SBP (mmHg)	111 ± 7.34	109 ± 8.87	110 ± 11.83	0.451
DBP (mmHg)	72 ± 7.63	72 ± 6.20	70 ± 8.73	0.415
**Dietary intake**				
Energy (kcal/day)	2770 ± 594.50	2705 ± 602.17	2596 ± 477.97	0.588
Carbohydrates (g/day)	257.43 ± 80.73	241.26 ± 73.92	246.74 ± 59.49	0.867
Sugar (g/day)	115.86 ± 34.83	111.65 ± 35.04	113.89 ± 41.02	0.906
Fiber (g/day)	45.17 ± 21.95	42.12 ± 14.65	38.93 ± 15.07	0.768
Protein (g/day)	103.72 ± 29.47	110.17 ± 31.86	107.75 ± 27.51	0.598
Total fat (g/day)	144.55 ± 29.17	141.83± 35.35	129.53 ± 28.96	0.249
SFAs (g/day)	37.61 ± 10.00	38.18 ± 11.04	36.81 ± 13.02	0.871
MUFAs (g/day)	70.37 ± 16.12	69.06 ± 17.17	59.46 ± 15.87	0.093
PUFAs (g/day)	25.91 ± 6.76	23.99 ± 7.25	23.59 ± 6.59	0.541
**Urinary phenolic metabolites (mg/day)**				
Lignans	26.63 ± 12.05	27.18 ± 7.19	29.01 ± 15.26	0.140
Hydroxybenzoic acids	56.05 ± 24.91	67.74 ± 59.66	71.71 ± 49.26	0.755
Hydroxycinnamic acids	2.93 ± 3.55	2.23 ± 1.99	2.17 ± 1.92	0.960
Stilbenes	0.49 ± 0.74	1.88 ± 1.90	1.82 ± 1.75	0974
Hydroxycoumarins	7.99 ± 5.93	7.26 ± 4.17	7.25 ± 5.84	0.732
**Eicosanoids**				
TXA_2_ (pg/mL)	1409 ± 31.96	1297 ± 65.81	1315 ± 53.55	0.673
PGI_2_ (pg/mL)	10,997 ± 57.57	10,495 ± 47.39	7927 ± 42.01	0.150
TXA_2_:PGI_2_ ratio	0.21 ± 0.19	0.14 ± 0.07	0.17 ± 0.10	0.681

Data are expressed as mean ± SD. CB: control butter; SRP: skin roasted peanuts; PB: peanut butter; BMI: body mass index; TG: triglyceride; TC: total cholesterol; LDL-c: LDL-cholesterol; HDL-c: HDL-cholesterol; DBP: diastolic blood pressure; SBP: systolic blood pressure; SFAs: saturated fatty acids; MUFAs: monounsaturated fatty acids; PUFAs: polyunsaturated fatty acids; PGI_2_: prostacyclin I2; TXA_2_: thromboxane A2. The p column refers to differences between groups at baseline; *p*-values < 0.05 are statistically significant and were calculated by the chi-square test for categorical variables and the Kruskal–Wallis test for continuous variables.

**Table 2 antioxidants-12-00698-t002:** Urinary phenolic metabolite concentrations in healthy young adults from the ARISTOTLE study before and after the intervention.

	SRP (n = 21)		PB (n = 22)		CB (n = 19)		*p*-Value	
	Pre- Intervention	Post- Intervention	Pre- Intervention	Post- Intervention	Pre- Intervention	Post- Intervention	SRP vs. CB	PB vs. CB
**Lignans-Lignans**	26.63 ± 12.05	34.17 ± 19.55	27.18 ± 7.19	35.60 ± 18.74	29.01 ± 15.26	26.36 ± 11.34	0.084	**0.038**
Enterodiol	18.26 ± 8.16	17.49 ± 8.42	19.89 ± 4.95	19.97 ± 7.51	19.75 ± 7.89	18.83 ± 7.95	0.836	0.824
Enterodiol glucuronide	1.78 ± 3.75	2.99 ± 3.07	0.83 ± 1.18	2.64 ± 5.07	1.47 ± 3.69	0.66 ± 0.97	**0.018**	**0.031**
Enterodiol sulfate	0.11 ± 0.14	0.23 ± 0.31	0.11 ± 0.11	0.11 ± 0.08	0.31 ± 0.65	0.29 ± 0.40	0.373	0.901
Enterolactone	0.27 ± 0.86	1.05 ± 3.63	0.09 ± 0.04	0.21 ± 0.26	0.10 ± 0.06	0.16 ± 0.23	0.206	0.545
Enterolactone glucuronide	6.11 ± 4.38	12.91 ± 11.99	6.17 ± 3.96	12.72 ± 10.04	7.27 ± 8.00	8.66 ± 10.78	**0.045**	**0.032**
Enterolactone diglucuronide	0.06 ± 0.04	0.05 ± 0.05	0.06 ± 0.03	0.05 ± 0.04	0.07 ± 0.05	0.08 ± 0.06	0.100	0.054
Enterolactone sulfate	0.03 ± 0.05	0.07 ± 0.05	0.04 ± 0.04	0.06 ± 0.04	0.05 ± 0.06	0.04 ± 0.05	**0.020**	0.086
**Hydroxybenzoic acids**	56.05 ± 24.91	76.87 ± 40.28	67.74 ± 59.66	99.87 ± 77.31	71.71 ± 49.26	71.51 ± 49.34	0.169	0.059
3-Hydroxybenzoic acid	3.95 ± 3.73	7.29 + 4.25	2.90 ± 1.72	6.46 ± 3.75	4.18 ± 2.00	3.87 ± 2.52	**0.002**	**<0.001**
4-Hydroxybenzoic acid	0.25 ± 0.40	0.69 ± 1.29	0.07 ± 0.07	0.38 ± 0.75	0.12 ± 0.18	0.39 ± 1.25	0.631	0.914
Hydroxybenzoic acid glucuronide	0.20 ± 0.23	0.53 ± 0.64	0.13 ± 0.16	0.37 ± 0.39	0.38 ± 0.58	0.57 ± 1.62	0.832	0.958
Hydroxybenzoic acid sulfate	3.17 ± 5.06	6.36 ± 4.04	3.83 ± 3.30	10.44 ± 7.21	4.69 ± 3.24	5.59 ± 6.43	0.090	**0.004**
Protocatechuic acid	1.13 ± 0.86	1.10 ± 1.14	2.08 ± 1.80	1.72 ± 1.71	1.30 ± 0.82	1.92 ± 3.18	0.331	0.335
Protocatechuic acid glucuronide I	0.72 ± 0.49	1.08 ±1.03	1.06 ± 1.18	1.44 ± 2.06	0.69 ± 0.47	0.63 ± 0.58	0.157	0.291
Protocatechuic acid glucuronide II	0.24 ± 0.11	0.19 ± 0.16	0.21 ± 0.19	0.25 ± 0.26	0.21 ± 0.17	0.20 ± 0.17	0.083	0.921
Protocatechuic acid sulfate	0.40 ± 0.20	1.55 ± 1.31	0.71 ± 0.73	2.58 ± 2.72	0.80 ± 0.75	2.82 ± 4.45	0.354	0.885
Vanillic acid	8.10 ± 6.02	6.63 ± 5.78	16.69 ± 25.82	13.69 ± 21.19	9.46 ± 12.23	10.27 ± 9.38	0.079	0.177
Vanillic acid glucuronide I	6.42 ± 3.92	5.52 ± 3.57	10.65 ± 16.93	10.24 ± 13.70	10.58 ± 14.43	10.52 ± 14.02	0.146	0.515
Vanillic acid glucuronide II	10.84 ± 9.96	11.55 ± 10.07	11.83 ± 12.70	18.90 ± 25.36	10.32 ±9.84	12.53 ± 12.50	0.112	0.976
Vanillic acid sulfate	12.96 ± 13.58	16.02 ± 18.58	12.13 ± 12.01	17.72 ± 14.94	16.50 ± 16.34	10.24 ± 11.58	**0.048**	**0.006**
Syringic acid	1.22 ± 1.35	2.56 ± 2.37	1.04 ± 0.16	1.79 ± 1.28	1.93 ± 1.84	2.07 ± 3.97	0.083	0.149
Syringic acid glucuronide I	1.14 ± 0.69	2.34 ± 3.25	0.93 ± 0.69	2.02 ± 2.95	1.08 ± 1.57	1.38 ± 2.07	0.267	0.165
Syringic acid glucuronide II	1.96 ± 2.24	2.35 ± 2.76	1.63 ± 2.78	2.52 ± 3.41	4.58 ± 6.83	1.28 ± 1.45	0.052	**0.023**
Syringic acid sulfate	3.33 ± 1.93	11.08 ± 9.85	4.85 ± 5.39	10.46 ± 6.09	4.88 ± 4.33	7.22 ± 10.23	**0.041**	0.208
**Hydroxycinnamic acids**	2.93 ± 3.55	3.75 ± 2.68	2.23 ± 1.99	5.47 ± 5.08	2.17 ± 1.92	1.29 ± 0.99	**0.040**	**0.001**
*p*-Coumaric acid	0.54 ± 1.34	0.37 ± 0.33	0.18 ± 0.21	0.43 ± 0.47	0.41 ± 0.74	0.17 ± 0.16	**0.046**	**0.016**
*m*-Coumaric acid	0.53 ± 0.95	0.36 ± 0.57	0.39 ± 1.12	0.40 ± 0.49	0.32 ± 0.38	0.35 ± 0.43	0.454	0.919
*o*-Coumaric acid	0.33 ± 0.84	0.12 ± 0.14	0.19 ± 0.22	0.22 ± 0.30	0.20 ± 0.18	0.15 ± 0.18	0.127	0.054
Coumaric acid glucuronide I	0.32 ± 0.83	0.48 ± 0.78	0.15 ± 0.17	0.33 ± 0.39	0.19 ± 0.34	0.12 ± 0.17	**0.001**	**0.040**
Coumaric acid glucuronide II	0.08 ± 0.18	0.15 ± 0.21	0.06 ± 0.05	0.13 ± 0.17	0.05 ± 0.06	0.06 ± 0.05	**0.003**	**0.036**
Coumaric acid glucuronide III	0.24 ± 0.28	0.43 ± 0.82	0.28 ± 0.46	0.67 ± 1.06	0.17 ± 0.15	0.28 ± 0.38	0.672	0.095
Coumaric acid glucuronide IV	0.12 ± 0.16	0.35 ± 0.49	0.22 ± 0.47	0.39 ± 0.45	0.10 ± 0.24	0.16 ± 0.22	0.673	0.493
Isoferulic acid	0.78 ± 0.64	1.49 ± 1.22 *	0.79 ± 0.53	2.95 ± 4.32 *	0.74 ± 1.08	0.02 ± 0.01	**0.013**	**0.015**
**Stilbenes**	2.04 ± 3.18	3.37 ± 4.86	1.88 ± 1.90	3.55 ± 3.84	1.82 ± 1.75	2.22 ± 3.59	0.371	0.882
Dihydroresveratrol glucuronide I	0.49 ± 0.74	1.17 ± 3.78	0.64 ± 1.18	1.42 ± 2.79	1.03 ± 1.34	0.66 ± 0.51	0.274	0.146
Dihydroresveratrol glucuronide II	1.26 ± 3.18	1.80 ± 2.95	0.98 ± 1.33	1.69 ± 2.07	0.56 ± 0.85	1.26 ± 3.50	0.068	**0.004**
Dihydroresveratrol glucuronide III	0.28 ± 0.19	0.38 ± 0.33	0.26 ± 0.14	0.43 ± 0.61	0.22 ± 0.17	0.30 ± 0.24	0.506	0.508
**Hydroxycoumarins**	7.99 ± 5.93	12.77 ± 14.88	7.26 ± 4.17	12.73 ± 10.14	7.25 ± 5.84	11.65 ± 21.63	0.791	0.694
Urolithin A	0.77 ± 1.68	0.13 ± 0.15	0.50 ± 1.69	0.21 ± 0.34	0.19 ± 0.40	0.08 ± 0.09	0.123	0.572
Urolithin A glucuronide	6.96 ± 5.67	12.24 ± 14.70	6.50 ± 3.92	12.06 ± 9.71	6.79 ± 5.82	11.25 ± 21.51	0.768	0.720
Urolithin A sulfate	0.16 ± 0.17	0.25 ± 0.17	0.22 ± 0.20	0.30 ± 0.29	0.25 ± 0.14	0.27 ± 0.26	0.268	0.466
Urolithin B	0.10 ± 0.16	0.15 ± 0.50	0.04 ± 0.06	0.20 ± 0.79	0.02 ± 0.03	0.04 ± 0.06	0.807	0.425

Data are expressed as mean (mg/day) ± SD. CB: control butter; SRP: skin roasted peanuts; PB: peanut butter. The *p*-value columns refer to differences adjusted by age and sex between SRP and PB vs. CB at 6 months and were calculated by a generalized estimating equation; *p*-values < 0.05 were considered significant.

**Table 3 antioxidants-12-00698-t003:** Urinary eicosanoid levels in healthy adults from the ARISTOTLE study before and after the intervention.

	SRP (n = 21)		PB (n = 22)		CB (n = 19)		*p*-Value	
	Pre- Intervention	Post- Intervention	Pre- Intervention	Post- Intervention	Pre- Intervention	Post- Intervention	SRP vs. CB	PB vs. CB
TXA_2_ (pg/mL)	1409 ± 31.96	1428 ± 81.76	1297 ± 65.81	1139 ± 53.39	1315 ± 53.55	1410 ± 59.59	0.456	0.414
PGI_2_ (pg/mL)	10,997 ± 57.57	14,607 ± 73.13	10,495 ± 47.39	13,773 ± 74.30	7927 ± 42.01	8548 ± 61.70	**0.037**	0.070
TXA_2_:PGI_2_ ratio	0.21 ± 0.19	0.13 ± 0.10	0.14 ± 0.07	0.10 ± 0.05	0.17 ± 0.10	0.22 ± 0.16	**0.008**	**0.047**

Data are expressed as mean (pg/mL) ± standard deviation (SD). SRP: skin roasted peanuts; PB: peanut butter; CB: control butter; PGI_2:_ prostacyclin I2; TXA_2_: thromboxane A2. The *p*-value columns refer to adjusted differences by age, sex, and physical activity between SRP and PB vs. CB at 6 months and were calculated by a generalized estimating equation; *p*-values < 0.05 were considered significant.

## Data Availability

Upon request, data described in the manuscript, code book, and analytic code will be made available.

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
