# Peer review of "Urinary Phenolic Metabolites Associated with Peanut Consumption May Have a Beneficial Impact on Vascular Health Biomarkers"

_antioxidants, 2023, doi:10.3390/antiox12030698_

Round 1

Reviewer 1 Report

Dear authors,

After the review process, I have several comments: in the abstract is necessary to include more numerical data; in the Materials and Methods sections should add references; in page 10, you should expand the discussion started from the bioavailability of phenolic compounds and microbiota bioactivity; the limitation of the study should be included, and the future valorization correlated with microbiota pattern modulation.

 Best regards!

Author Response

We sincerely thank the editor and all reviewers for the opportunity to have our manuscript reviewed in Antioxidants, and for their valuable suggestions. 

Please see the attachment with the revisions made. 

Reviewer 2 Report

This is a classical randomized human trial intervention.

Consequently, results are not always as clear as we should expect. Nevertheless, in this particular case, the authors have performed a very robust study to ensure a high degree of trust on the final results. 

I just have a very small amount of minor typo corrections I found when I was reading this interesting work.

Line 244 – correct Urolothin A to urolithin A.

Line 273 – correct glucuronide to glucuronide

Along the text, correct the subscript of TXA2, PGI2

Author Response

(The authors gave the same response as above.)

Reviewer 3 Report

Thank you for giving me possibility to prepare review for sending manuscript with title:  Urinary phenolic metabolites associated with peanut consumption may have a beneficial impact on vascular health biomarkers

 Manuscript is very interesting and well written. The title is exactly proper with topic presented in manuscript.

 The Abstract. In this sub-section authors give a short presentation of manuscript. This section is well constructed. Please add some representative values for main results to Abstract section, to make easier for reading.

 Keywords: It would be good to arrange keywords in alphabetical order. Can you to change it?

 Introduction section.

 The Introduction section contains all the necessary information related to the presented topic of the article.

Page 1, lines 44-45: I understand, that polyphenols are a very large group of bioactive compounds, but please give more examples of phenolics (characteristic for peanuts butter) from different chemical groups: for example per one or two from different groups: flavonoids….; phenolic acids…., stilbens:….. and would be good to add some values for those compounds, to show what rich source of those compounds are peanuts.

At the end of Introduction section, a properly formulated research aim is included.

Materials and method section

 This section is construct well. I am impressed about works done by Authors.

 Results.

In my opinion results that are more valuable would be, when Authors add table with chemical analysis of peanut butter (with polyphenols analysis)

 All Tables and Figures are legible and well structured. The authors correctly included the statistical tools.

The discussion section presents a good comparison of the obtained results with other results available in the data basis.

Presented conclusions are corresponding with all information presented via Authors’ in manuscript text. A very interesting addition to the Conclusions chapter are the strengths and weaknesses of the thesis. This shows the practical message of the study.

Final opinion:  After carefully manuscript reading, I think that presented manuscript is a very valuable and it could be publish in Antioxidants journal after correction according to my suggestions.

Author Response

(The authors gave the same response as above.)
